# On the Recursive Teaching Dimension
# of VC Classes

**Xi Chen**
Department of Computer Science
Columbia University
xichen@cs.columbia.edu

**Yu Cheng**
Department of Computer Science
University of Southern California
yu.cheng.1@usc.edu

**Bo Tang**
Department of Computer Science
Oxford University
tangbonk1@gmail.com

## Abstract

The *recursive teaching dimension* (RTD) of a concept class $\mathcal{C} \subseteq \{0, 1\}^n$, introduced by Zilles et al. [ZLHZ11], is a complexity parameter measured by the worst-case number of labeled examples needed to learn any target concept of $\mathcal{C}$ in the *recursive teaching model*. In this paper, we study the quantitative relation between RTD and the well-known learning complexity measure VC dimension (VCD), and improve the best known upper and (worst-case) lower bounds on the recursive teaching dimension with respect to the VC dimension.

Given a concept class $\mathcal{C} \subseteq \{0, 1\}^n$ with $\text{VCD}(\mathcal{C}) = d$, we first show that $\text{RTD}(\mathcal{C})$ is at most $d \cdot 2^{d+1}$. This is the first upper bound for $\text{RTD}(\mathcal{C})$ that depends only on $\text{VCD}(\mathcal{C})$, independent of the size of the concept class $|\mathcal{C}|$ and its domain size $n$. Before our work, the best known upper bound for $\text{RTD}(\mathcal{C})$ is $O(d2^d \log \log |\mathcal{C}|)$, obtained by Moran et al. [MSWY15]. We remove the $\log \log |\mathcal{C}|$ factor.

We also improve the lower bound on the worst-case ratio of $\text{RTD}(\mathcal{C})$ to $\text{VCD}(\mathcal{C})$. We present a family of classes $\{\mathcal{C}_k\}_{k \geq 1}$ with $\text{VCD}(\mathcal{C}_k) = 3k$ and $\text{RTD}(\mathcal{C}_k) = 5k$, which implies that the ratio of $\text{RTD}(\mathcal{C})$ to $\text{VCD}(\mathcal{C})$ in the worst case can be as large as $5/3$. Before our work, the largest ratio known was $3/2$ as obtained by Kuhlmann [Kuh99]. Since then, no finite concept class $\mathcal{C}$ has been known to satisfy $\text{RTD}(\mathcal{C}) > (3/2) \cdot \text{VCD}(\mathcal{C})$.

## 1 Introduction

In computational learning theory, one of the fundamental challenges is to understand how different *information complexity* measures arising from different learning models relate to each other. These complexity measures determine the worst-case number of labeled examples required to learn any concept from a given concept class. For example, one of the most notable results along this line of research is that the sample complexity in PAC-learning is linearly related to the VC dimension [BEHW89]. Recall that the VC dimension of a concept class $\mathcal{C} \subseteq \{0, 1\}^n$ [VC71], denoted by $\text{VCD}(\mathcal{C})$, is the maximum size of a *shattered* subset of $[n] = \{1, \dots, n\}$, where we say $Y \subseteq [n]$ is shattered if for every binary string $\boldsymbol{b}$ of length $|Y|$, there is a concept $c \in \mathcal{C}$ such that $c|_Y = \boldsymbol{b}$. Here we use $c|_X$ to denote the projection of $c$ on $X$. As the best-studied information complexity measure, VC dimension is known to be closely related to many other complexity parameters, and it serves as a natural parameter to compare against across various models of learning and teaching.

Instead of the PAC-learning model where the algorithm takes random samples, we consider an interactive learning model where a helpful teacher selects representative examples and present them to the learner, with the objective of minimizing the number of examples needed. The notion of a *teaching set* was introduced in mathematical models for teaching. The teaching set of a concept $c \in \mathcal{C}$ is a set of indices (or examples) $X \subseteq [n]$ that uniquely identifies $c$ from $\mathcal{C}$. Formally, given a concept class $\mathcal{C} \subseteq \{0,1\}^n$ (a set of binary strings of length $n$), $X \subseteq [n]$ is a *teaching set* for a concept $c \in \mathcal{C}$ (a binary string in $\mathcal{C}$) if $X$ satisfies

$$c|_X \neq c'|_X, \quad \text{for all other concepts } c' \in \mathcal{C}.$$

The *teaching dimension* of a concept class $\mathcal{C}$ is the smallest number $t$ such that *every* $c \in \mathcal{C}$ has a teaching set of size no more than $t$ [GK95, SM90]. However, teaching dimension does not always capture the cooperation in teaching and learning (as we will see in Example 2), and a more optimistic and realistic notion of *recursive teaching dimension* has been introduced and studied extensively in the literature [Kuh99, DSZ10, ZLHZ11, WY12, DFSZ14, SSYZ14, MSWY15].

**Definition 1.** The recursive teaching dimension of a class $\mathcal{C} \subseteq \{0,1\}^n$, denoted by $\text{RTD}(\mathcal{C})$, is the smallest number $t$ where one can order all the concepts of $\mathcal{C}$ as an ordered sequence $c_1, \ldots, c_{|\mathcal{C}|}$ such that every concept $c_i$, $i < |\mathcal{C}|$, has a teaching set of size no more than $t$ in $\{c_i, \ldots, c_{|\mathcal{C}|}\}$.

Hence, $\text{RTD}(\mathcal{C})$ measures the worst-case number of labeled examples needed to learn any target concept in $\mathcal{C}$, if the teacher and the learner are cooperative. We would like to emphasize that an optimal ordered sequence (as in Definition 1) can be derived by the teacher and learner separately without any communication: They can put all concepts in $\mathcal{C}$ that have the smallest teaching dimension appear at the beginning of the sequence, then remove these concepts from $\mathcal{C}$ and proceeds recursively. By definition, $\text{RTD}(\mathcal{C})$ is always bounded from above by the teaching dimension of $\mathcal{C}$ but can be much smaller than the teaching dimension. We use the following example to illustrate the difference between the teaching dimension and the *recursive* teaching dimension.

**Example 2.** Consider the class $\mathcal{C} \subseteq \{0,1\}^n$ with $n+1$ concepts: the empty concept $\mathbf{0}$ and all the singletons. For example when $n = 3$, $\mathcal{C} = \{000, 100, 010, 001\}$. Each singleton concept has teaching dimension 1, while the teaching dimension for the empty concept $\mathbf{0}$ is $n$, because the teacher has to reveal all labels to distinguish $\mathbf{0}$ from the other concepts. However, if the teacher and the learner are cooperative, every concept can be taught with one label: If the teacher reveals a "0" label, the learner can safely assume that the target concept must be $\mathbf{0}$, because otherwise the teacher would present a "1" label instead for the other concepts. Equivalently, in the setting of Definition 1, the teacher and the learner can order the concepts so that the singleton concepts appear before the empty concept $\mathbf{0}$. Then every concept has a teaching set of size 1 to distinguish it from the later concepts in the sequence, and thus the recursive teaching dimension of $\mathcal{C}$ is 1.

In this paper, we study the quantitative relationship between the recursive teaching dimension (RTD) and the VC dimension (VCD). A bound on the RTD that depends only on the VCD would imply a close connection between learning from random samples and teaching (under the recursive teaching model). The same structural properties that make a concept class easy to learn would also give a bound on the number of examples needed to teach it. Moreover, the recursive teaching dimension is known to be closely related to sample compression schemes [LW86, War03, DKSZ16], and a better understanding of the relationship between RTD and VCD might help resolve the long-standing sample compression conjecture [War03], which states that every concept class has a sample compression scheme of size linear in its VCD.

## 1.1 Our Results

Our main result (Theorem 3) is an upper bound of $d \cdot 2^{d+1}$ on $\text{RTD}(\mathcal{C})$ when $\text{VCD}(\mathcal{C}) = d$. This is the first upper bound for $\text{RTD}(\mathcal{C})$ that depends only on $\text{VCD}(\mathcal{C})$, but not on $|\mathcal{C}|$, the size of the concept class, or $n$, the domain size. Previously, Moran et al. [MSWY15] showed an upper bound of $O(d2^d \log \log |\mathcal{C}|)$ for $\text{RTD}(\mathcal{C})$; our result removes the $\log \log |\mathcal{C}|$ factor, and answers positively an open problem posed in [MSWY15].

Our proof tries to reveal examples iteratively to minimize the number of the remaining concepts. Given a concept class $\mathcal{C} \subseteq \{0,1\}^n$, we pick a set of examples $Y \subseteq [n]$ and their labels $\boldsymbol{b} \in \{0,1\}^Y$, so that the set of remaining concepts (with the projection $c|_Y = \boldsymbol{b}$) is *nonempty* and has the *smallest* size among all choices of $Y$ and $\boldsymbol{b}$. We then prove by contradiction (with the assumption of $\text{VCD}(\mathcal{C}) = d$)

that, if the size of $Y$ is large enough (but still depends on only $\text{VCD}(\mathcal{C})$), the remaining concepts must have VC dimension at most $d - 1$. This procedure gives us a recursive formula, which we solve and obtain the claimed upper bound on RTD of classes of VC dimension $d$.

We also improve the lower bound on the worst-case factor by which RTD may exceed VCD. We present a family of classes $\{\mathcal{C}_k\}_{k \geq 1}$ (Figure 4) with $\text{VCD}(\mathcal{C}_k) = 3k$ and $\text{RTD}(\mathcal{C}_k) = 5k$, which shows that the worst-case ratio between $\text{RTD}(\mathcal{C})$ and $\text{VCD}(\mathcal{C})$ is at least $5/3$. Before our work, the largest known multiplicative gap between $\text{RTD}(\mathcal{C})$ and $\text{VCD}(\mathcal{C})$ was a ratio of $3/2$, given by Kuhlmann [Kuh99]. (Later Doliwa et al. [DFSZ14] showed the smallest class $\mathcal{C}_W$ with $\text{RTD}(\mathcal{C}_W) = (3/2) \cdot \text{VCD}(\mathcal{C}_W)$ (Warmuth's class)). Since then, no finite concept class $\mathcal{C}$ with $\text{RTD}(\mathcal{C}) > (3/2) \cdot \text{VCD}(\mathcal{C})$ has been found.

Instead of exhaustively searching through all small concept classes, our improvement on the lower bound is achieved by formulating the existence of a concept class with the desired RTD, VCD and domain size, as a boolean satisfiability problem. We then run the state-of-the-art SAT solvers on these formulae to discover a concept class $\mathcal{C}_0$ with $\text{VCD}(\mathcal{C}_0) = 3$ and $\text{RTD}(\mathcal{C}_0) = 5$. Based on the concept class $\mathcal{C}_0$, one can produce a family of concept classes $\{\mathcal{C}_k\}_{k \geq 1}$ with $\text{VCD}(\mathcal{C}_k) = 3k$ and $\text{RTD}(\mathcal{C}_k) = 5k$, by taking the Cartesian product of $k$ copies of $\mathcal{C}_0$: $\mathcal{C}_k = \mathcal{C}_0 \times \ldots \times \mathcal{C}_0$.

## 2   Upper Bound on the Recursive Teaching Dimension

In this section, we prove the following upper bound on $\text{RTD}(\mathcal{C})$ with respect to $\text{VCD}(\mathcal{C})$.

**Theorem 3.** *Let $\mathcal{C} \subseteq \{0,1\}^n$ be a class with $\text{VCD}(\mathcal{C}) = d$. Then $\text{RTD}(\mathcal{C}) \leq 2^{d+1}(d-2) + d + 4$.*

Given a class $\mathcal{C}$, we use $\text{TS}_{\min}(\mathcal{C})$ to denote the smallest integer $t$ such that *at least one* concept $c \in \mathcal{C}$ has a teaching set of size $t$. Notice that $\text{TS}_{\min}(\mathcal{C})$ is different from teaching dimension. Teaching dimension is defined as the smallest $t$ such that *every* $c \in \mathcal{C}$ has a teaching set of size at most $t$.) Theorem 3 follows directly from Lemma 4 and the observation that the VC dimension of a concept class does not increase after a concept is removed. (After removing a concept from $\mathcal{C}$, the new class $\mathcal{C}'$ still has $\text{VCD}(\mathcal{C}') \leq d$, and one can apply Lemma 4 again to obtain another concept that has a teaching set of the desired size in $\mathcal{C}'$ and repeat this process.)

**Lemma 4.** *Let $\mathcal{C} \subseteq \{0,1\}^n$ be a class with $\text{VCD}(\mathcal{C}) = d$. Then $\text{TS}_{\min}(\mathcal{C}) \leq 2^{d+1}(d-2) + d + 4$.*

We start with some intuition by reviewing the proof of Kuhlmann [Kuh99] that every class $\mathcal{C}$ with $\text{VCD}(\mathcal{C}) = 1$ must have a concept $c \in \mathcal{C}$ with a teaching set of size 1. Given an index $i \in [n]$ and a bit $b \in \{0,1\}$, we use $\mathcal{C}_b^i$ to denote the set of concepts $c \in \mathcal{C}$ such that $c_i = b$. The proof starts by picking an index $i$ and a bit $b$ such that $\mathcal{C}_b^i$ is *nonempty* and has the *smallest* size among all choices of $i$ and $b$. The proof then proceeds to show that $\mathcal{C}_b^i$ contains a unique concept, which by the definition of $\mathcal{C}_b^i$ has a teaching set $\{i\}$ of size 1. To see why $\mathcal{C}_b^i$ must be a singleton set, we assume for contradiction that it contains more than one concept. Then there exists an index $j \neq i$ and two concepts $c, c' \in \mathcal{C}_b^i$ such that $c_j = 0$ and $c'_j = 1$. Since $\mathcal{C}$ has $\text{VCD}(\mathcal{C}) = 1$, $\{i, j\}$ cannot be shattered and thus, all the concepts $c^* \in \mathcal{C}$ with $c_i^* = 1 - b$ must share the same $c_j^*$, say $c_j^* = 0$. As a result, it is easy to verify that $\mathcal{C}_1^j$ is a nonempty proper subset of $\mathcal{C}_b^i$, contradicting the choice of $i$ and $b$ at the beginning.

Moran et al. [MSWY15] used a similar approach to show that every so-called $(3,6)$-class $\mathcal{C}$ has $\text{TS}_{\min}(\mathcal{C})$ at most 3. They define a class $\mathcal{C} \subseteq \{0,1\}^n$ to be a $(3,6)$-class if for any three indices $i, j, k \in [n]$, the projection of $\mathcal{C}$ onto $\{i, j, k\}$ has at most 6 patterns. (In contrast, $\text{VCD}(\mathcal{C}) = 2$ means that the projection of $\mathcal{C}$ has at most 7 patterns. So $\mathcal{C}$ being a $(3,6)$-class is a stronger condition than $\text{VCD}(\mathcal{C}) = 2$.) The proof of [MSWY15] starts by picking two indices $i, j \in [n]$ and two bits $b_1, b_2 \in \{0,1\}$ such that $\mathcal{C}_{b_1,b_2}^{i,j}$, i.e., the set of $c \in \mathcal{C}$ such that $c_i = b_1$ and $c_j = b_2$, is *nonempty* and has the *smallest* size among all choices of $i, j$ and $b_1, b_2$. They then prove by contradiction that $\text{VCD}(\mathcal{C}_{b_1,b_2}^{i,j}) = 1$, and combine with [Kuh99] to conclude that $\text{TS}_{\min}(\mathcal{C}) \leq 3$.

Our proof extends this approach further. Given a concept class $\mathcal{C} \subseteq \{0,1\}^n$ with $\text{VCD}(\mathcal{C}) = d$, let $k = 2^d(d-1) + 1$ and we pick a set $Y^* \subset [n]$ of $k$ indices and a string $\boldsymbol{b}^* \in \{0,1\}^k$ such that $\mathcal{C}_{\boldsymbol{b}^*}^{Y^*}$, the set of $c \in \mathcal{C}$ such that the projection $c|_{Y^*} = \boldsymbol{b}^*$, is *nonempty* and has the *smallest* size among all choices of $Y$ and $\boldsymbol{b}$. We then prove by contradiction (with the assumption of $\text{VCD}(\mathcal{C}) = d$) that $\mathcal{C}_{\boldsymbol{b}^*}^{Y^*}$ must have VC dimension at most $d - 1$. This gives us a recursive formula that bounds the $\text{TS}_{\min}$ of classes of VC dimension $d$, which we solve to obtain the upper bound stated in Lemma 4.

We now prove Lemma 4.

*Proof of Lemma 4.* We prove by induction on $d$. Let
$$f(d) = \max_{\mathcal{C}: \text{VCD}(\mathcal{C}) \le d} \text{TS}_{\min}(\mathcal{C}).$$
Our goal is to prove the following upper bound for $f(d)$:
$$f(d) \le 2^{d+1}(d-2) + d + 4, \quad \text{for all } d \ge 1. \tag{1}$$
The base case of $d = 1$ follows directly from [Kuh99].

For the induction step, we show that condition (1) holds for some $d > 1$, assuming that it holds for $d - 1$. Take any concept class $\mathcal{C} \subseteq \{0,1\}^n$ with $\text{VCD}(\mathcal{C}) \le d$. Let $k = 2^d(d-1) + 1$. If $n \le k$ then we are already done because
$$\text{TS}_{\min}(\mathcal{C}) \le n \le k = 2^d(d-1) + 1 \le 2^{d+1}(d-2) + d + 4,$$
where the last inequality holds for all $d \ge 1$. Assume in the rest of the proof that $n > k$. Then any set of $k$ indices $Y \subset [n]$ partitions $\mathcal{C}$ into $2^k$ (possibly empty) subsets, denoted by
$$\mathcal{C}_{\boldsymbol{b}}^Y = \{c \in \mathcal{C} : c|_Y = \boldsymbol{b}\}, \quad \text{for each } \boldsymbol{b} \in \{0,1\}^k.$$
We follow the approach of [Kuh99] and [MSWY15] to choose a set of $k$ indices $Y^* \subset [n]$ as well as a string $\boldsymbol{b}^* \in \{0,1\}^k$ such that $\mathcal{C}_{\boldsymbol{b}^*}^{Y^*}$ is *nonempty* and has the *smallest* size among all nonempty $\mathcal{C}_{\boldsymbol{b}}^Y$, over all choices of $Y$ and $\boldsymbol{b}$. Without loss of generality we assume below that $Y^* = [k]$ and $\boldsymbol{b}^* = \boldsymbol{0}$ is the all-zero string. For notational convenience, we also write $\mathcal{C}_{\boldsymbol{b}}$ to denote $\mathcal{C}_{\boldsymbol{b}}^{Y^*}$ for $\boldsymbol{b} \in \{0,1\}^k$.

Notice that if $\mathcal{C}_{\boldsymbol{b}^*} = \mathcal{C}_{\boldsymbol{b}^*}^{Y^*}$ has VC dimension at most $d - 1$, then we have
$$\text{TS}_{\min}(\mathcal{C}) \le k + f(d-1) \le 2^{d+1}(d-2) + d + 4,$$
using the inductive hypothesis. This is because according to the definition of $f$, one of the concepts $c \in \mathcal{C}_{\boldsymbol{b}^*}$ has a teaching set $T \subseteq [n] \setminus Y^*$ of size at most $f(d-1)$ to distinguish it from other concepts of $\mathcal{C}_{\boldsymbol{b}^*}$. Thus, $[k] \cup T$ is a teaching set of $c$ in the original class $\mathcal{C}$, of size at most $k + f(d-1)$.

Figure 1: An illustration for the proof of Lemma 4, $\text{TS}_{\min}(\mathcal{C}) \le 6$ when $d = 2$. We prove by contradiction that the smallest nonempty set $\mathcal{C}_{\boldsymbol{b}^*}^{Y^*}$, after fixing five bits, has $\text{VCD}(\mathcal{C}_{\boldsymbol{b}^*}^{Y^*}) = 1$, where $Y^* = \{1,2,3,4,5\}$ and $\boldsymbol{b}^* = \boldsymbol{0}$. In this example, we have $Z = \{6,7\}$, $Y' = \{2,3,4,6,7\}$ and $\boldsymbol{b}' = \boldsymbol{0}$. Note that $\mathcal{C}_{\boldsymbol{0}}^{Y'}$ is indeed a nonempty proper subset of $\mathcal{C}_{\boldsymbol{0}}^{Y^*}$.

Finally, we prove by contradiction that $\mathcal{C}_{\boldsymbol{b}^*}$ has VC dimension at most $d - 1$. Assume that $\mathcal{C}_{\boldsymbol{b}^*}$ has VC dimension $d$. Then by definition there exists a set $Z \subseteq [n] \setminus Y^*$ of $d$ indices that is shattered by $\mathcal{C}_{\boldsymbol{b}^*}$ (i.e., all the $2^d$ possible strings appear in $\mathcal{C}_{\boldsymbol{b}^*}$ on $Z$). Observe that for each $i \in Y^*$, the union of all $\mathcal{C}_{\boldsymbol{b}}$ with $b_i = 1$ (recall that $\boldsymbol{b}^*$ is the all-zero string) must miss at least one string on $Z$, which we denote by $\boldsymbol{p}_i$ (choose one arbitrarily if more than one are missing); otherwise, $\mathcal{C}$ has a shattered set of size $d + 1$, i.e., $Z \cup \{i\}$, contradicting with the assumption that $\text{VCD}(\mathcal{C}) \le d$. (See Figure 1 for an example when $d = 2$ and $k = 5$.) However, given that there are only $2^d$ possibilities for each $\boldsymbol{p}_i$ (and $|Y^*| = k = 2^d(d-1) + 1$), it follows from the pigeonhole principle that there exists a subset $K \subset Y^*$ of size $d$ such that $\boldsymbol{p}_i = \boldsymbol{p}$ for every $i \in K$, for some $\boldsymbol{p} \in \{0,1\}^d$. Let $Y' = (Y^* \setminus K) \cup Z$ be a new set of $k$ indices and let $\boldsymbol{b}' = \boldsymbol{0}_{k-d} \circ \boldsymbol{p}$. Then $\mathcal{C}_{\boldsymbol{b}'}^{Y'}$ is a nonempty and proper subset of $\mathcal{C}_{\boldsymbol{b}^*}^{Y^*}$, a contradiction with our choice of $Y^*$ and $\boldsymbol{b}^*$.

This finishes the induction and the proof of the lemma. □

# 3 Lower Bound on the Worst-Case Recursive Teaching Dimension

We also improve the lower bound on the worst-case factor by which RTD may exceed VCD.

In this section, we present an improved lower bound on the worst-case factor by which $\text{RTD}(\mathcal{C})$ may exceed $\text{VCD}(\mathcal{C})$. Recall the definition of $\text{TS}_{\min}(\mathcal{C})$, which denotes the number of examples needed to teach *some* concept in $c \in \mathcal{C}$. By definition we always have $\text{RTD}(\mathcal{C}) \geq \text{TS}_{\min}(\mathcal{C})$ for any class $\mathcal{C}$.

Kuhlmann [Kuh99] first found a class $\mathcal{C}$ such that $\text{RTD}(\mathcal{C}) = \text{TS}_{\min}(\mathcal{C}) = 3$ and $\text{VCD}(\mathcal{C}) = 2$, with domain size $n = 16$ and $|\mathcal{C}| = 24$. Since then, no class $\mathcal{C}$ with $\text{RTD}(\mathcal{C}) > (3/2) \cdot \text{VCD}(\mathcal{C})$ has been found. Recently, Doliwa et al. [DFSZ14] gave the smallest such class $\mathcal{C}_W$ (Warmuth's class, as shown in Figure 2), with $\text{RTD}(\mathcal{C}_W) = \text{TS}_{\min}(\mathcal{C}_W) = 3$, $\text{VCD}(\mathcal{C}_W) = 2$, $n = 5$, and $|\mathcal{C}_W| = 10$. We can view $\mathcal{C}_W$ as taking all five possible rotations of the two concepts $(0, 0, 0, 1, 1)$ and $(0, 1, 0, 1, 1)$.

| $x_1$ | $x_2$ | $x_3$ | $x_4$ | $x_5$ |
|---|---|---|---|---|
| 0 | 0 | 0 | 1 | 1 |
| 0 | 0 | 1 | 1 | 0 |
| 0 | 1 | 1 | 0 | 0 |
| 1 | 1 | 0 | 0 | 0 |
| 1 | 0 | 0 | 0 | 1 |
| 0 | 1 | 0 | 1 | 1 |
| 1 | 0 | 1 | 1 | 0 |
| 0 | 1 | 1 | 0 | 1 |
| 1 | 1 | 0 | 1 | 0 |
| 1 | 0 | 1 | 0 | 1 |

(a)

| $x_1$ | $x_2$ | $x_3$ | $x_4$ | $x_5$ |
|---|---|---|---|---|
| 0 | 0 | 0 | 1 | 1 |
| 0 | 1 | 0 | 1 | 1 |

(b)

Figure 2: (a) Warmuth's class $\mathcal{C}_W$ with $\text{RTD}(\mathcal{C}_W) = 3$ and $\text{VCD}(\mathcal{C}_W) = 2$; (b) The succinct representation of $\mathcal{C}_W$ with one concept selected from each rotation-equivalent set of concepts. The teaching set of each concept is marked with underline.

Given $\mathcal{C}_W$ one can obtain a family of classes $\{\mathcal{C}_k\}_{k \geq 1}$ by taking the Cartesian product of $k$ copies:

$$\mathcal{C}_k = \mathcal{C}_W^k = \mathcal{C}_W \times \cdots \times \mathcal{C}_W,$$

and it follows from the next lemma that $\text{RTD}(\mathcal{C}_k) = \text{TS}_{\min}(\mathcal{C}_k) = 3k$ and $\text{VCD}(\mathcal{C}_k) = 2k$.

**Lemma 5** (Lemma 16 of [DFSZ14]). *Given two concept classes $\mathcal{C}_1$ and $\mathcal{C}_2$.*

*Let $\mathcal{C}_1 \times \mathcal{C}_2 = \{(c_1, c_2) \mid c_1 \in \mathcal{C}_1, c_2 \in \mathcal{C}_2\}$. Then*

$$\begin{aligned}
\text{TS}_{\min}(\mathcal{C}_1 \times \mathcal{C}_2) &= \text{TS}_{\min}(\mathcal{C}_1) + \text{TS}_{\min}(\mathcal{C}_2), \\
\text{RTD}(\mathcal{C}_1 \times \mathcal{C}_2) &\leq \text{RTD}(\mathcal{C}_1) + \text{RTD}(\mathcal{C}_2), \quad \textit{and} \\
\text{VCD}(\mathcal{C}_1 \times \mathcal{C}_2) &= \text{VCD}(\mathcal{C}_1) + \text{VCD}(\mathcal{C}_2).
\end{aligned}$$

Lemma 5 allows us to focus on finding small concept classes with $\text{RTD}(\mathcal{C}) > (3/2) \cdot \text{VCD}(\mathcal{C})$. The first attempt to find such classes is to exhaustively search over all possible binary matrices and then compute and compare their VCD and RTD. But brute-force search quickly becomes infeasible as the domain size $n$ gets larger. For example, even the class $\mathcal{C}_W$ has fifty $0/1$ entries. Instead, we formulate the existence of a class with certain desired RTD, VCD, and domain size, as a boolean satisfiability problem, and then run state-of-the-art Boolean Satisfiability (SAT) solvers to see whether the boolean formula is satisfiable or not.

We briefly describe how to construct an equivalent boolean formula in conjunctive normal form (CNF). For a fixed domain size $n$, we have $2^n$ basic variables $x_c$, each describing whether a concept $c \in \{0, 1\}^n$ is included in $\mathcal{C}$ or not. We need VC dimension to be at most VCD, which is equivalent to requiring that every set $S \subseteq [n]$ of size $|S| = \text{VCD} + 1$ is not shattered by $\mathcal{C}$. So we define auxiliary variables $y_{(S,b)}$ for each set $S$ of size $|S| = \text{VCD} + 1$, and every string $\boldsymbol{b} \in \{0, 1\}^S$, indicating whether a specific pattern $\boldsymbol{b}$ appears in the projection of $\mathcal{C}$ on $S$ or not. These auxiliary variables are decided by the basic variables, and for every $S$, at least one of the $2^{|S|}$ patterns must be missing on $S$.

For the minimum teaching dimension to be at least RTD, we cannot teach any row with RTD − 1 labels. So for every concept $c$, and every set of indices $T \subseteq [n]$ of size $|T| = \text{RTD} - 1$, we need at least one other concept $c' \neq c$ satisfying $c|_T = c'|_T$ so that $c'$ is there to "confuse" $c$ on $T$. As an example, we list one clause of each type, from the SAT instance with $n = 5$, VCD = 2, and RTD = 3:

$$x_{01011} \to y_{(\{1,2,3\},010)}, \quad \bigvee_b \neg\, y_{(\{1,2,3\},b)}, \quad x_{01011} \to \bigvee_{b \neq 011} x_{(01,b)}.$$

Note that there are many ways to formulate our problem as a SAT instance. For example, we could directly use a boolean variable for each entry of the matrix. But in our experiments, the SAT solvers run faster using the formulation described above. The SAT solvers we use are Lingeling [Bie15] and Glucose [AS14] (based on MiniSAT [ES03]). We are able to rediscover $\mathcal{C}_W$ and rule out the existence of concept classes for certain small values of $(\text{VCD}, \text{RTD}, n)$; see Figure 3.

| VCD($\mathcal{C}$) | RTD($\mathcal{C}$) | $n$ (domain size) | Satisfiable | Concept Class |
|---|---|---|---|---|
| 2 | 3 | 5 | Yes | $\mathcal{C}_W$ (Figure 2) |
| 2 | 4 | 7 | No | |
| 3 | 5 | 7 | No | |
| 3 | 6 | 8 | No | |
| 4 | 6 | 7 | No | |
| 4 | 7 | 8 | No | |
| 3 | 5 | 12 | Yes | Figure 4 |

Figure 3: The satisfiability of the boolean formulae for small values of VCD($\mathcal{C}$), RTD($\mathcal{C}$), and $n$.

Unfortunately for $n > 8$, even these SAT solvers are no longer feasible. We use another heuristic to speed up the SAT solvers when we conjecture the formula to be satisfiable — adding additional clauses to the SAT formula so that it has fewer solutions (but hopefully still satisfiable), and faster to solve. More specifically, we bundle all the rotation-equivalent concepts, that is if we include a concept, we must also include all its rotations. Note that with this restriction, we can reduce the number of variables by having one for each rotation-equivalent set; we can also reduce the number of clauses, since if $S$ is not shattered, then we know all rotations of $S$ are also not shattered.

We manage to find a class $\mathcal{C}_0$ with $\text{RTD}(\mathcal{C}_0) = \text{TS}_{\min}(\mathcal{C}_0) = 5$ and VCD($\mathcal{C}$) = 3, and domain size $n = 12$. A succinct representation of $\mathcal{C}_0$ is given in Figure 4, where all rotation-equivalent concepts (i.e. rows) are omitted. The first 8 rows each represents 12 concepts, and the last row represents 4 concepts (because it is more symmetric), with a total of $|\mathcal{C}_0| = 100$ concepts. We also include a text file with the entire concept class $\mathcal{C}_0$ (as a $100 \times 12$ matrix) in the supplemental material. Applying Lemma 5, we obtain a family of concept classes $\{\mathcal{C}_k\}_{k \geq 1}$, where $\mathcal{C}_k = \mathcal{C}_0 \times \cdots \times \mathcal{C}_0$ is the Cartesian product of $k$ copies of $\mathcal{C}_0$, that satisfy $\text{RTD}(\mathcal{C}_k) = 5k$ and VCD($\mathcal{C}_k$) = 3k.

## 4    Conclusion and Open Problem

We improve the best known upper and lower bounds for the worst-case recursive teaching dimension with respect to VC dimension. Given a concept class $\mathcal{C}$ with $d = \text{VCD}(\mathcal{C})$ we improve the upper bound $\text{RTD}(\mathcal{C}) = O(d2^d \log\log |\mathcal{C}|)$ of Moran et al. [MSWY15] to $2^{d+1}(d - 2) + d + 4$, removing the $\log\log |\mathcal{C}|$ factor as well as the dependency on $|\mathcal{C}|$. In addition, we improve the lower bound $\max_{\mathcal{C}}(\text{RTD}(\mathcal{C})/\text{VCD}(\mathcal{C})) \geq 3/2$ of Kuhlmann [Kuh99] to $\max_{\mathcal{C}}(\text{RTD}(\mathcal{C})/\text{VCD}(\mathcal{C})) \geq 5/3$.

Our results are a step towards answering the following question:

$$\textit{Is } \text{RTD}(\mathcal{C}) = O(\text{VCD}(\mathcal{C}))?$$

posed by Simon and Zilles [SZ15].

While Kuhlmann [Kuh99] showed that $\text{RTD}(\mathcal{C}) = 1$ when $\text{VCD}(\mathcal{C}) = 1$, the simplest case that is still open is to give a tight bound on $\text{RTD}(\mathcal{C})$ when $\text{VCD}(\mathcal{C}) = 2$: Doliwa et al. [DFSZ14] presented a concept class $\mathcal{C}$ (Warmuth's class) with $\text{RTD}(\mathcal{C}) = 3$, while our Theorem 3 shows that $\text{RTD}(\mathcal{C}) \leq 6$.

| $x_1$ | $x_2$ | $x_3$ | $x_4$ | $x_5$ | $x_6$ | $x_7$ | $x_8$ | $x_9$ | $x_{10}$ | $x_{11}$ | $x_{12}$ |
|---|---|---|---|---|---|---|---|---|---|---|---|
| 0 | 0 | 0 | 0 | 0 | 1 | 0 | 1 | 0 | 1 | 0 | 1 |
| 0 | 0 | 0 | 0 | 0 | 1 | 1 | 1 | 0 | 1 | 0 | 1 |
| 0 | 0 | 0 | 0 | 1 | 1 | 0 | 1 | 0 | 1 | 0 | 1 |
| 0 | 0 | 0 | 1 | 0 | 1 | 1 | 1 | 0 | 1 | 0 | 1 |
| 0 | 0 | 0 | 1 | 1 | 1 | 0 | 1 | 0 | 1 | 0 | 1 |
| 0 | 0 | 1 | 1 | 0 | 1 | 0 | 1 | 0 | 1 | 0 | 1 |
| 0 | 0 | 1 | 1 | 0 | 1 | 1 | 1 | 0 | 1 | 0 | 1 |
| 0 | 1 | 0 | 1 | 0 | 1 | 1 | 1 | 0 | 1 | 1 | 1 |
| 0 | 1 | 1 | 1 | 0 | 1 | 1 | 1 | 0 | 1 | 1 | 1 |

Figure 4: The succinct representation of a concept class $\mathcal{C}_0$ with $\mathrm{RTD}(\mathcal{C}_0) = 5$ and $\mathrm{VCD}(\mathcal{C}_0) = 3$. The teaching set of each concept is marked with underline.

**Acknowledgments**

We thank the anonymous reviewers for their helpful comments and suggestions. We also thank Joseph Bebel for pointing us to the SAT solvers. This work was done in part while the authors were visiting the Simons Institute for the Theory of Computing. Xi Chen is supported by NSF grants CCF-1149257 and CCF-1423100. Yu Cheng is supported in part by Shang-Hua Teng's Simons Investigator Award. Bo Tang is supported by ERC grant 321171.

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
