[Reviews · NeurIPS 2016]

Reviewer 1

Summary

The paper studies the connection between recursive teaching dimension (RTD) of a concept class and the VC dimension. They provide an upper bound of d*2^d for the RTD of a class with CV dimension d using a clever argument that generalizes previous work of Moran et al and Kuhlmann et al. Notably, this upper bound does not have a dependency on the size of the concept class. Using an explicit construction they show that RTC >= 5/3VC in the worst case.

Qualitative Assessment

Overall the arguments in the paper are elegant and build upon previous work well. I have two main comments: 1. Even though teaching dimension is a fairly abstract and theoretical concept, it would be good to motivate the ideas through some applications if possible. The lack of applications and the fact that the result is purely theoretical is what motivates my rating of 2 for usefulness. I do think the presentation is extremely clear and the paper very easy to read motivating the rating of 3. For example the authors can provide references to other works that use recursive teaching dimension in applications. 2. The question of how RTD is related to VC is interesting and the results of this paper still contains an exponential gap between these quantities. Because this question and the result in this paper is aimed at such a narrow audience, the paper might be a better fit for a conference like COLT rather than NIPS. 3. In general the teaching dimension has no real connection to the VC dimension. This is why the conjecture stating that RTD = O(VC) is so interesting. However there is recent work (Ji Liu, Xiaojin Zhu, and H. Gorune Ohannessian. The Teaching Dimension of Linear Learners) that establishes the teaching dimension of several types of linear classes. Though the setting studied in this paper is more discrete, it would be interesting to understand what the recursive teaching dimension of those classes is and how it relates to the vc dimension in those cases (assuming this question is well posed). This paper may be better suited for a conference like COLT, where related work has appeared.

Confidence in this Review

2-Confident (read it all; understood it all reasonably well)


Reviewer 2

Summary

The paper is the first to provide an upper bound on the recursive teaching dimension (a complexity parameter in co-operative teacher/learner settings) in terms of the VC-dimension that does not depend on the size of the concept class. The upper bound is exponential in the VC-dimension. Further, the paper is the first to show that there are concept classes whose recursive teaching dimension exceeds their VC-dimension by more than a factor of 3/2 (namely, they provide classes that exhibit a factor of 5/3).

Qualitative Assessment

The paper is very insightful - the authors quite nicely explain the approach they took for proving their results. The questions addressed, while interesting only for a fairly small subcommunity of the machine learning community, are really important in that subcommunity, and the authors have achieved a substantial breakthrough on an open problem posed in COLT 2015. I quite liked the idea to formulate the problem of finding a concept class with RTD > 3/2 VCD as a SAT problem. In my eyes, the results should definitely be published, and they are important enough to deserve publication in a leading venue like NIPS. The paper is generally well written and easy to read, but there are a few minor (easy to fix issues (mostly just typos etc). The two things I'd like the authors to fix, since they are rather misleading, are the following: (1) They keep talking about proving a "lower bound" on RTD in terms of VCD, and even in the abstract they say "RTD(C)\ge (5/3)VCD(C) in general". While I know what they want to say, what they do say is actually incorrect. There are classes of RTD 1 that have arbitrarily large VCD, so there is no lower bound on RTD in terms of VCD, and certainly (5/3)VCD(C) is not a lower bound. But 5/3 is a lower bound on the worst-case factor by which RTD may exceed VCD. The authors need to go through the whole paper and fix this everywhere (I spotted it here: last para of abstract, heading of section 3, first sentence of section 3, line 1 on page 7, line 2 on page 7, but I may have overlooked other places). (2) After Definition 1, the authors say that "RTD(C) measures the worst-case number of labeled examples needed to learn any target concept in C, if the teacher and the learner agree a priori on a specific order of the concepts". While this is not incorrect, is is rather misleading, since no such a priori agreement is needed. If teacher and learner do not agree on an order, they can still use a well-defined process to uniquely derive teaching sets, and the largest teaching set will have a size equal to RTD(C): first, one determines ALL concepts in C that have the smallest teaching dimension in C, and assigns teaching sets accordingly. Then one removes ALL those concepts from the class and proceeds recursively. Hence, it would create the wrong impression to say that RTD corresponds to some order that the two parties have to agree on a priori. I'd like to urge the authors to modify the writing in this point. Everything else is just typos and the like: line 9: at most d2^{d+1} -- > in O(d2^{d+1}) line 11: is -- > was line 16: is -- > was line 22: resultS line 83: largest _known_ multiplicative ... line 84: is -- > was lines 116 and 118: I believe it should be VCD(C) = 2 in both cases (not 3) formula display after line 134: VCD should be TS_min same after line 141 line 181: delete "that" at the end of the line line 206: insert a "times" symbol before the last C_0

Confidence in this Review

3-Expert (read the paper in detail, know the area, quite certain of my opinion)


Reviewer 3

Summary

This paper studies the relationship between the VC dimension of a concept class C and its recursive teaching dimension (RTD). Prior work: -- If the class is "maximal" then VC = RTD -- In general, RTD = O(VC 2^{VC} log log |C|) This paper: RTD = O(VC 2^{VC}) In addition, the paper provides an example of a concept class for which RTD > 1.66 VC.

Qualitative Assessment

This is a nice result with a clever proof. It uses ideas of recent work by Moran et al, and answers one of their open questions. Although the upper bound seems to only improve by a log log factor, the removal of the dependence on the concept class size is significant. The lower bound is uninteresting -- in fact, it is found using a SAT solver.

Confidence in this Review

3-Expert (read the paper in detail, know the area, quite certain of my opinion)


Reviewer 4

Summary

The main idea in this paper is to extend the results in [MSWY15] and [Kuh99] to construct a lower and upper bound on recursive teaching dimension (RTD) that depends only on the VC dimension (VCD) to elucidate the connection between learning from random samples and teaching.

Qualitative Assessment

The proof of the upper bound (Lemma 4) is constructed by a "prove by contradiction" method using the pigeon-hole principle on lines 145-155. The other portions of this upper bound proof are identical to those presented in [MSWY15] and [Kuh99]. To construct the lower bound the authors construct a boolean satisfiability problem for specific RTD, VCD, and domain size to detect whether the constraints in (Lemma 5) from [DFSZ14] can be satisfied or not. The main novelty of the paper appears to be contained on lines 145-155 and lines 177-192. However, these do not provide a significant technical extension to the results in [MSWY15] and [Kuh99].

Confidence in this Review

2-Confident (read it all; understood it all reasonably well)


Reviewer 5

Summary

The paper provides new upper and lower bounds on Recursive Teaching Dimension (RTD) of a concept class given only its VC dimension (VCD). The upper bound is (d.(2^{d+1}) and thus removing the dependence on the size of the concept class from the bound obtained by Moran et al. in 2015 which was (O(d.(2^{d}).log log |C|)) where d is the VCD and |C| is the size of the concept class. The lower bound obtained is ((5/3).d) which is tighter than the previous one obtained by kuhlmann in 1999 (3/2).d. The upper bound proof extended the approach of Moran et al. in a recursive way and used induction. For the lower bound, a SAT solver was used after modeling the problem in boolean formulas where they obtained a concept class with RTD = 5 and VCD = 3 where a cartesian product can be taken of copies of this class to obtain classes with RTD = 5k and VCD = 3k. The paper is a step to solve the open problem posed by Simon and Zilles in 2015 which is: Is RTD(C) = O(VCD(C))?

Qualitative Assessment

The paper is a theoretical paper which provides relations between the VC dimension, resembling the complexity of a concept class given a PAC-learning algorithm where the examples are given at random, and the Recursive Teaching Dimension, where examples are chosen by a teacher and the order of concepts is agreed-on by the learner and the teacher. The bounds are an enhanced version of previous bounds. The proof for the upper bound seems to be sound. However, for the lower bound, I did not get how the statement in line 83 "... which implies that RTD(C) >= (5/3).VCD(C) in general" is correct although there is the Warmuth's class which has RTD=(3/2).VCD and Kuhlmann proof in 1999 that if a concept class has a VCD=1 then RTD=1. I cannot see it is a lower bound as claimed in lines 87-88. Moreover, In lines 116-117 it shouldn't be "... VCD(C) = 3 mean that the projection of C has at most 8 patterns" (rather than 7)? The presentation is clear and the paper is nice to read. I think it has valuable contribution to learning theory and as stated might help resolve the sample compression conjecture by M. K. Warmuth in 2003.

Confidence in this Review

2-Confident (read it all; understood it all reasonably well)


Reviewer 6

Summary

The paper demonstrates that RTD(H) = O(2^(VC-dim(H))). this improves over previous results by Moran et al. that showed bounds that are sample size dependent. Thus this paper is the first to show that RTD(H) is bounded by the VC dimension solely. Other lower bounds are considered.

Qualitative Assessment

Given the audience of NIPS it is hard to give this paper an Oral-level score. However, the paper is original, well motivated and solves a non-trivial problem in Learning Theory. I find it completely within the scope. The proof is elegant, simple and uses fundamental tools. Some suggestion: Lately Shay Moran showed that compression schemes also have a strikingly similar bound. It is noteworthy that whenever a bound for compression scheme appear a similar bound appears for RTD. It is worth mentioning that currently the two problems (both strongly related to VC-dim) enjoy the same bound. ***within the discussion period it has come to my attention the formulating the open problem not carefully as I formulated it is wrong.

Confidence in this Review

3-Expert (read the paper in detail, know the area, quite certain of my opinion)